# A Linear Model to Describe Branching and Allometry in Root Architecture

**DOI:** 10.3390/plants8070218

**Published:** 2019-07-12

**Authors:** Joel Colchado-López, R. Cristian Cervantes, Ulises Rosas

**Affiliations:** Jardín Botánico, Instituto de Biología, Universidad Nacional Autónoma de México, 04510 Mexico City, Mexico

**Keywords:** *Arabidopsis thaliana*, allometric modeling, linear model, natural variation, root architecture, root branching, phenotyping

## Abstract

Root architecture is a complex structure that comprises multiple traits of the root phenotype. Novel platforms and models have been developed to better understand root architecture. In this methods paper, we introduce a novel allometric model, named rhizochron index (*m*), which describes lateral root (LR) branching and elongation patterns across the primary root (PR). To test our model, we obtained data from 16 natural accessions of *Arabidopsis thaliana* at three stages of early root development to measure conventional traits of root architecture (e.g., PR and LR length), and extracted the rhizochron index (*m*). In addition, we tested previously published datasets to assess the utility of the rhizochron index (*m*) to distinguish mutants and environmental effects on root architecture. Our results indicate that rhizochron index (*m*) is useful to distinguish the natural variations of root architecture between *A. thaliana* accessions, but not across early stages of root development. Correlation analyses in these accessions showed that *m* is a novel trait that partially captures information from other root architecture traits such as total lateral root length, and the ratio between lateral root and primary root lengths. Moreover, we found that the rhizochron index was useful to distinguish ABA effect on root architecture, as well as the mutant *pho1* phenotype. We propose the rhizochron index (*m*) as a new feature of the root architectural system to be considered, in addition to conventional traits in future investigations.

## 1. Introduction

Root architecture is a concept that involves the spatial arrangement of the root network within a substrate, encompassing the different kinds of roots, their developmental origins and functional specializations, and the growth dynamics of the system. The root architectural structure takes into account topological, geometrical, and morphological features which provide fractions of the whole picture around a root phenotype [1,2]. Conventionally, some traits like primary root length, lateral root density, or the presence of root-derived structures (e.g., root hair presence) have often been used to describe the root architecture and have been reviewed elsewhere [3]. Nevertheless, these traits by themselves only account for local characteristics of root architecture. Thus, models that allow for the integration of multiple traits are necessary to understand root architecture systems as a whole.

In *Arabidopsis thaliana*, as in many plant species, the root grows at the tip where the primary meristem is located. In this area of the root, precise cell divisions and cell elongations occur to originate the tissue layers, including the epidermis, cortex, the endodermis, and the stele. As cell divisions and cell elongations pushes cells away from the root tip, cell populations mainly elongate to constitute what is known as the elongation zone of the root. Once the cells reach a certain size and position away from the metistematic zone, the differentiation zone is visible by the presence of root hairs. The differentiation zone is also the place where the outer layer of the stele, the pericycle meristem, acquires competence to develop lateral root primordia, some of which elongate to become visible lateral roots, replicating the basic anatomy, meristematic activity, elongation, and differentiation patterns of the primary root. Thus, the root architecture structure is composed of the primary root, plus the lateral root branches, and their corresponding lateral root branches, all of which can be measured in several conventional ways such as lengths. Thus, root architecture is the result of several developmental events, which display diversity in natural variants, such as different rates of primary root growth, lateral root formation, or ratios of root proliferation, as it has been seen in *A. thaliana* accessions, or developmental plasticity due to environmental conditions such as phosphorous deficit. 

To bridge the gap in understanding of root architecture using conventional traits, novel ways to quantify roots have been developed in recent years. These techniques include new methodological strategies for root phenotyping [3,4], improved imaging analysis platforms focused on root systems that increase efficiency and reproducibility [5,6,7,8], and novel complex models that unveil root dynamics [9,10]. Some of these novel approaches integrate allometric models to more precisely capture the architectural systems of plant organs, without prior assumptions of what features of the root could be biologically relevant. In other words, novel multivariate models of architectural traits might uncover cryptotypes, which are allometric relationships between organs or sections of the same organ. More importantly, these cryptotypes might be potentially important because it is likely that they are the target of natural selection, and therefore there might be genetic bases underlying them [11] rather than the usually considered conventional traits [12]. In other words, natural selection is more likely to see organs and organisms as a whole or as the interaction of their components rather than their isolates units. As proxy to understand organs and organisms, quantitative characterizations of unidimensional features have been highly valuable. However, we need to explore novel ways to further capture the complexity of organs and organisms using multitrait approaches, aiming at understanding what is really important in nature.

While observing *A. thaliana* roots, it is easily noted that longer lateral roots are often located closer to the proximal end of the primary root, and shorter lateral roots are located towards the distal end of the root. This is because the primary root does not branch close to the tip, but in the differentiation zone. This pattern could be captured as a function of the lateral root length as dependent of its position along the primary root. Little attention has been paid to this pattern as a potential proxy of the whole root architecture. This is why we believed that analyzing natural accessions, mutants, and environmental conditions could yield some insights into the relevance of the lateral root patterns along the primary root. 

Here we propose a new method to characterize root system architecture, using the lengths of lateral roots and the branching pattern along the primary root in *A. thaliana*. In order to study natural variation in developmental changes in root architecture, we characterized root phenotypes in a set of *A. thaliana* accessions and a dataset of previously published root phenotypes. To obtain a new root phenotype, we propose a linear function named the “rhizochron index (*m*)” which describes both how often the elongated lateral root emerges, and the length of the elongated lateral roots. The rhizochron index (*m*) was then compared to the conventional phenotypes to figure out what aspects of root development this new phene captures.

## 2. Results

To characterize natural variation in root architecture, we grew 16 accessions on vertical plates and scanned them at 8, 10, and 12 days after sowing. We calculated the primary root length and the total length of elongated lateral roots. This showed the variation on root phenotypes between accessions and across developmental time points (Figure 1) as has been shown before [11,13].

In root architectural systems, primary roots originate early in development, and lateral roots branch thereafter. Thus, it is expected that primary root length is longer than lateral root length if both elongate at similar rates. In our data, we observed this pattern where primary root length is generally larger than lateral root length, and rates of growth are similar following parallel elongation trajectories (Figure 1). This is the case of Edi-0 and other accessions such as Rmx-A02, and Spr1-2. However, in most accessions growth rates are less parallel: Total lateral root elongates at a higher rate than primary root (Ga-0), or the primary root elongates at a higher rate than total lateral root length (Rmx-A180). Remarkably, we observed a tendency where the rate of lateral root growth surpassed the rate of primary root growth by 12 days after germination. This was the case for many accessions, such as Col-0 and Shahdara, and in one instance, this overtake occurred earlier at day 10 (Wei-0).

Parametrization of root architecture. Several traits can be used to describe aspects of root architecture, but none of them by themselves are able to provide an integrative picture of how the *Arabidopsis* root develops. However, we noticed that often the most rootward elongated lateral roots tend to be shorter than the most shootward counterparts. This is because often rootward lateral roots are younger than shootward lateral roots. As lateral root initiation and elongation are discrete and temporally successive, yet relatively delayed, processes, there should often occur a negative relationship between the distance from the shoot to the elongated lateral root and the length of the lateral root itself. In other words, the further away from the shoot, the shorter the elongated lateral root [14]. This can be expressed as a linear model y = mx + b, where y is the length of the elongated lateral root and x is the position of the lateral root along the primary root. A linear model can be fitted to any root having at least two elongated lateral roots, corresponding to the coordinates x_1_, y_1_ for the most shootward lateral root, and x_2_, y_2_ for the most rootward lateral root (Figure 2). This was the reason why the accession Rmx-A108 was discarded from the analysis—because the elongation of lateral roots up to day 12 was poor and irregular (Figure 1, Appendix A). Other individuals per accession were also discarded because of the lack of more than one elongated lateral root (Appendix A).

We calculated the slope (*m*) on the linear regression of each individual at each developmental stage. *m* was defined as the rhizochron index and then used as a trait to statistically compare the rhizochron distribution patterns, taking together the replicates from each of the accessions and each of the developmental time-points (Figure 3). We observed that *m* values exhibit different degrees of variation, with some accessions showing little changes through developmental stages (e.g., Edi-0), and some others displaying tendencies towards slightly higher *m* values as development progresses, as in C24 and Spr1-6. To evaluate the significance of our observations, we performed a two-way ANOVA with interaction to test for accession (15 levels) and developmental stage (3 levels) effects, and found that developmental stage does not explain the variation (F_(2,798)_ = 0.881; *p* = 0.415) nor the interaction of accession and developmental stage (F_(28, 798)_ = 0.591; *p* = 0.955), as shown in Figure 3. This indicates that though we observed in some accessions the tendency of median-*m* value to subtly change over time (e.g., C24 and Spr1-6), the effect of this variation does not account for global effects on the rhizochron index by developmental stages. However, we found that genetic background does significantly affect the rhizochron index (F_(14,798)_ = 4.769; *p* < 0.001). In other words, the rhizochron index (*m*) has the power to differentiate accessions, but not developmental stages.

To confirm the previous observation, we performed a one-way ANOVA to test for the accession effect on the rhizochron index (*m*) per accession at all stages of development, or at day 12. When testing *m* at all stages (Figure 4A), we found that accession had a significant effect (F_(14,828)_ = 4.866; *p* < 0.001). A post-hoc Tukey test (HSD) showed differences between accessions, grouping them within seven statistically homogenous groups. Further, we noticed that accession-dependent effects were slightly more significant when using exclusively data from day 12 (F_(14, 281)_ = 5.346, *p* < 0.001), also showing seven statistically homogeneous groups (Tukey test for HSD, Figure 4B). This indicates that the rhizochron index (*m*) is more suitable to distinguish variation in root architecture between accessions at later stages rather than earlier stages of development. 

Up to now, we have shown the usefulness of the rhizochron index (*m*) to distinguish natural *A. thaliana* accessions, but we wondered whether this new trait was also useful to detect phenotypic differences in root architecture on plants exposed to different environments. To do this, we obtained image data from previously published researches [11] and calculated the rhizochron index (*m*) from those datasets. In the first one the effect of the plant hormone and phosphorous (P) deficiency was tested in *A. thaliana* accession Col-0 [11]. When re-measuring the root image data, we confirmed the previous results reported for primary root length (PRL) and total lateral root length (TLRL). A two-way ANOVA for ABA (F_(1,188)_ = 0.128, *p* = 0.721) and P (F_(1,188)_ = 1.9, *p* = 0.17), showed that there are no significant effects on the PRL (Figure 5A). However, there were significant effects of ABA (F_(1,188)_ = 98.8, *p* < 0.001) and P (F_(1,188)_ = 5.79, *p* = 0.02) on the TLRL (Figure 5B). To reassess this data under the light of the rhizochron index (*m*), we obtained the *m* values for each of the root images. The corresponding two-way ANOVA showed that ABA had a significant effect on *m* (F_(1,188)_ = 153.92, *p* < 0.001), although no significant effects were detected on P (F_(1,188)_ = 1.34, *p* = 0.25) (Figure 5C). This shows that some phenotypic effects on root architecture can be detected using the rhizochron index (*m*). 

Natural accessions can be considered to be natural mutants; however, we also wondered whether the rhizochron index (*m*) is useful to evaluate artificial mutants. Thus, we used another dataset from the previously published research [11], where they tested the phenotype of the mutant for the phosphate transporter *pho1* in *A. thaliana* accession Col-0 [15]. As expected, a two-tailed t-test between the wild type (WT) and *pho1* showed significant differences on PRL (t_(36.64)_ = −2.48, *p* = 0.02) and TLRL (t_(22.10)_ = −7.77, *p* < 0.001) (Figure 5D–E). Consistent with these results, we also obtained significant differences on the rhizochron index (*m*) between the WT and *pho1* (t_(37.98)_ = 2.03, *p* = 0.048) (Figure 5F).

To test how the rhizochron index is related to conventional traits (e.g., primary root length, lateral root length, root density, etc.), we performed a Pearson correlation analysis on the dataset of natural accessions. We found that *m* is a root architecture estimate that correlates significantly with several conventionally measured attributes (Table 1). The highest correlation of *m* was found with the ratio between lateral and total root length (TLr) at day 8 (r^2^ = 0.149, *p* < 0.001). The second highest correlation was found with the ratio between lateral and primary root length (PLr), also at day 8 (r^2^ = 0.137, *p* < 0.001). Several other correlations were highly significant (*p* < 0.001), but the correlation coefficients were small, indicating that the rhizochron index does partially contain some of the conventionally measured attributes, but the conventional traits weakly predict the parameter *m* in the rhizochron index. 

## 3. Discussion

Root architecture in plants, and particularly in *A. thaliana* is complex, and has been the subject of conceptual and empirical discussion regarding the most efficient ways to capture their features [16,17]. We measured root architectural features to build a linear model that explains the patterns of lateral root branching, which allowed us to extract the parameter rhizochron index (*m*) as a descriptor of each root. We found that *m* is rather relevant to describe the natural variation of the root architecture at later stages of development. We also found that *m* is useful to distinguish environmental effects on root architecture and mutant phenotypes. Moreover, we found that *m* is a novel trait of root architecture that partially captures information from conventionally used traits, such as total lateral root length and the ratio between LR and PR, potentially representing a new cryptotype of root architecture [12].

Our findings show that rhizochron index (*m*) is a helpful trait to describe root architecture variation across natural accessions in *A. thaliana* under standard conditions, mainly at later stages of root development. In other words, it allows the numerical distinction of root phenotypes. As previously shown, the environment can largely modify root architecture [18]. Thus, we predict that *m* could also be useful to differentiate root architecture under different growth conditions, as we have shown for the presence of the plant hormone ABA, even though *m* was not able to capture the effect of phosphorous (P) deficiency (Figure 5C). It remains to be tested whether *m* can capture the effect of other environmental conditions on root architecture [19,20,21]. Moreover, the rhizochron index (*m*) was designed to evaluate a root architecture composed of a main root that has lateral branches, such as in *A. thaliana*. We believe *m* could be a particularly useful metric to describe the roots of other species with similar architectural patterns, such as soybean or oilseed rape.

Though our findings show that rhizochron index (*m*) varies across population, whether this natural variation has a genetic basis remains to be tested. Our reassessment of previously published phenotypes of the phosphate transport mutant *pho1* [11,15] allows us to conclude that the phenotype *m* can have genetic basis, as was the case for *pho1* (Figure 5F). As *m* seems to capture information of phenes (*sensu* York [22]) like PR length, LR length, and lateral branching, it is possible that the rhizochron index is a useful proxy to capture lateral root branching patterns, such as those found before using allometric estimations [7,23,24]. However, as we observed, *m* is weakly correlated to any of the conventionally quantified root architectural features (i.e., LRL, LPr, BZL), suggesting that *m* indeed is a novel trait, with overlapping underlying genetic architecture with two or more root architectural traits, though this hypothesis and its biological significance are yet to be elucidated.

Despite the potential of the rhizochron index (*m*) to capture root architecture phenotypes, we recognize that it might have pitfalls as an accurate descriptor of root architecture. First, we do not distinguish between lateral roots and basal roots. As in several other studies, basal roots are often measured together with lateral roots, although their developmental origin is different. Basal roots originate from a meristematic tissue within the transition zone and the hypocotyl. Therefore, basal roots might have a distinctive growth dynamic. Here, basal roots and lateral roots were measured indistinctly because of the difficulty to distinguish one from the other at the earlier stages of development and the transition zone between the shoot and the root as we relied exclusively on an image database. Second, the post-BZ length, which is the section of PR between the last visible and elongated lateral root, and the tip of PR, are not captured by the model, because *m* mainly focuses on the lateral root development within the branching zone, and by this definition the post-BZ is neglected. Further improvements of the model would be ideal to capture the post-BZ length. Third, the BZ is defined according to the elongated visible lateral roots, which leaves us potentially omitting non-elongated lateral root primordia. We are not suggesting that our model is a profound revolution on root system architecture phenotyping, but rather a useful bidimensional parameter that can distinguish between accessions and environments through the PR and LR elongation allometric relationship, and therefore, *m* could correspond to a new root cryptotype: A composite phenotype that is actually subject to natural selection, but has been overlooked. 

Notwithstanding the pitfalls of our model, rhizochron index (*m*) is still a novel trait that resumes the topology of LR branching and the geometry of LR and PR length into a single quantitative value, possibly a cryptotype [12]. Improving our knowledge of root architecture includes searching for novel ways to examine the phenotype of this organ, which may be further employed to dissect the genetic basis of root architecture. Then, this new information can be applied to study the environmental and developmental dynamics of root architecture and its effects on plant fitness, thus improving our understanding about which traits of root architecture are subjected to natural selection [25,26,27]. We argue that developing new models based on the allometric exploration of organs and organisms, such as *m* does, are at least as important as studying each trait individually due to the possibility that natural selection may not act directly on unidimensional features but rather on the multidimensional features of an organism’s phenotype and how these allometric relationships affect the overall set of responses an organism has to its environment. Therefore, any new allometric relationship that may putatively represent a biological cryptotype is to be considered as a useful parameter to study the fitness and adaptive value of any given phenotype. In line with these statements, as the information given by *m* values is only partially captured by conventionally used traits, we believe the rhizochron index (*m*) can be used as a proxy that brings to light new allometric information relevant to better the understanding of root architecture and its complexity.

## 4. Materials and Methods 

### 4.1. Plant Material

We used the accessions: C24 (CS22620, n = 20), Col-0 (n = 17), Edi-0 (CS22657, n = 21), Ga-0 (CS22634, n = 21), Mrk-0 (CS22635, n = 20), Ren-1 (CS22610, n = 21), Rmx-A02 (CS22568, n = 18) Rmx-A180 (CS22569, n = 22), Shahdara (CS22652, n = 21), Spr1-2 (CS22582, n = 20), Spr1-6 (CS22583, n = 19), Ull2-3 (CS22587, n = 21), Wa-1 (CS22644, n = 20), Wei-0 (CS22622, n = 17), Yo-0 (CS22624, n = 20), and Zdr-6 (CS22588, n = 21), obtained from the ABRC Stock Center, and propagated for one generation before performing the experiments. A subset of these plants was used for fitting the linear regression model, adjusted as is described in Figure 2. 

### 4.2. Growth Conditions

We washed the seeds in a solution of ethanol, commercial bleach, and water for 2–3 min, plus three rinses of sterile water. We sowed the seeds on square plates with MS media without sucrose and nitrogen (M5524, Sigma, St Louis, MO, USA), supplemented with 0.1% sucrose, 5 mM of KNO_3_, 0.05% MES salts (Gibco BRL, Gaithersburg, MD, USA), and 1% agar (Bacto Agar BD, Lakes, NJ, USA). After sowing, plates were maintained horizontally in the dark at 4 °C for 3 days for imbibition and priming of the seeds. Then, they were placed horizontally to a growth chamber (Intellus Percival Scientific, Perry, IA, USA) at 22 °C, long day (16 h light–8 h dark), and 125 μmol⋅m^−2^s^−1^ light intensity. 

### 4.3. Imaging and Phenotype Quantification

We obtained 300-dpi resolution images of the plates 8, 10, and 12 days after plates were transferred to the 22 °C chamber, using a V600 scanner (Epson). We measured digital images using the free software ImageJ. Root phenotype was quantified as described in Figure 2B, first measuring PR lengths in between each LR, followed by measuring each LR in a shoot-to-root-apex sense. R (Core Team) was used to compile the measurements obtained from ImageJ into data frames, and also to quantify the values for conventional features of root architecture and *m*, and to perform parametric statistics and fitted regressions.

### 4.4. Statistical Analyses

The statistical analyses shown in this manuscript were performed by using an R 3.5.1 (Core Team) platform. Two-way ANOVA analysis with interaction, t-test, and fitting linear regression were performed using the functions aov(), t.test(), and lm() from the base package. HSD.test() function from the agricolae package was used to perform post-hoc Tukey tests. Pearson’s correlation was done by using the rcorr() function from the Hmisc package.

## Figures and Tables

**Figure 1 plants-08-00218-f001:**
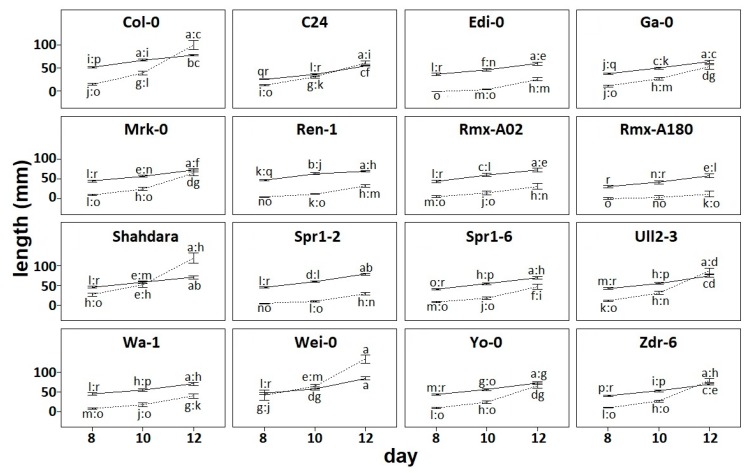
Root developmental variation across a set of *A. thaliana* accessions. Solid line indicates primary root length; dotted line indicates lateral root length. Error bars indicate standard error from the mean. Primary root length and total lateral root length were separately evaluated with a two-way ANOVA with interaction for accession (16 levels) and developmental stage (3 levels). Letters indicate post-hoc Tukey-HSD groups (*p* < 0.05) on each of the two traits. Full data available on Appendix A.

**Figure 2 plants-08-00218-f002:**
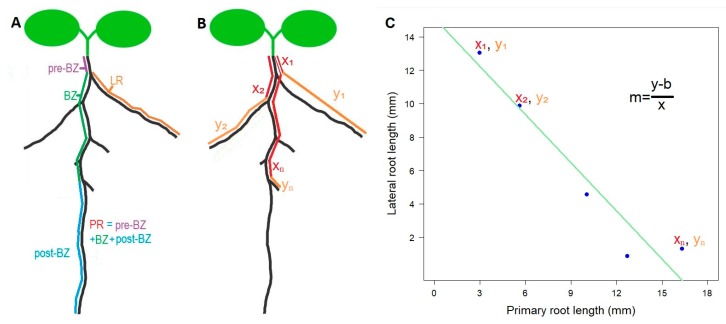
Construction of the rhizochron index (*m*) using an adjusted linear model. (**A**) Definition of conventional root architecture traits: pre-branching zone (pre-BZ) is the section between the shoot first elongated and visible lateral root (LR) along the primary root (PR), BZ (branching zone) is the section of the primary root with visible and elongated LR, and post-branching zone is the section of the PR from the last visible and elongated LR and the tip of the PR. (**B**) Root features obtained to estimate the rhizochron index in any given root. x_1_ is the distance between the shoot and the first LR along the PR, y_1_ is the length of the first LR, x_2_ is the distance between the shoot and the second LR along the PR, y_2_ is the length of the second LR, x_n_ is the distance between the shoot and the last LR along the PR, y_n_ is the length of the last LR. (**C**) Fitting a linear model using the estimates x_1_y_1_, x_2_y_2_, x_n_y_n_, and extracting the rhizochron index (*m*).

**Figure 3 plants-08-00218-f003:**
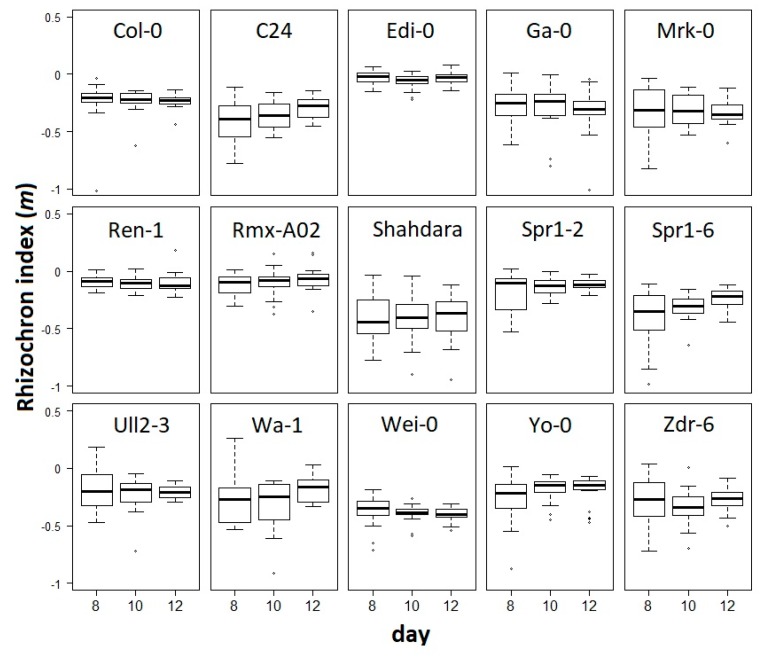
The rhizochron index (*m*) in a set of accessions during development. A two-way ANOVA with interaction was performed on *m* for accession (15 levels) and developmental stage (3 levels). Boxplots indicate median (line) and SE (whiskers) values.

**Figure 4 plants-08-00218-f004:**
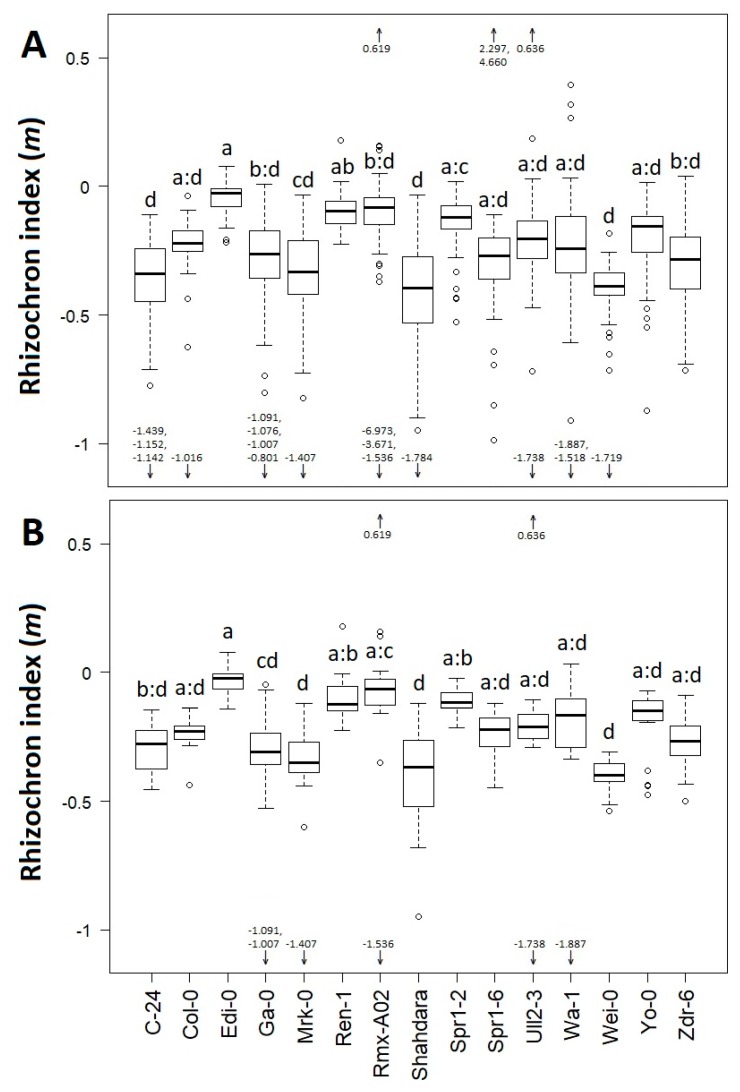
Variation of rhizochron index (*m*) across accessions. (**A**) Rhizochron index per accession taking together all stages of development. (**B**) Rhizochron index per accession taking only day 12. Letters indicate Tukey-HSD groups (*p* < 0.05) from a one-way ANOVA for accession effects (15 levels). Groups noted as ‘x:z’ mean ‘xyz’; for example, ‘a:c’ and ‘b:d’ mean ‘abc’ and ‘bcd’, respectively. Arrows indicate outliers above 0.5 or below −1.0. Boxplots indicate median (line) and SE (whiskers) values.

**Figure 5 plants-08-00218-f005:**
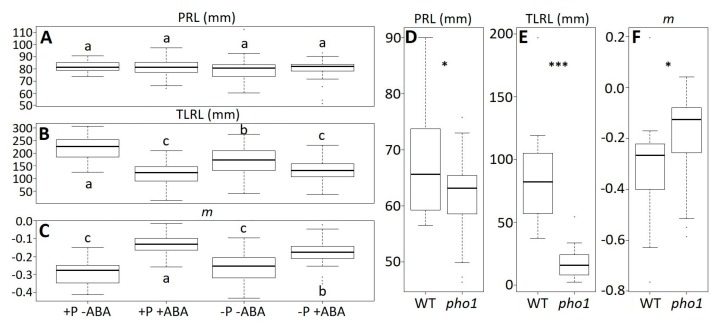
Environmental and mutant genotypic effects on rhizochron index (*m*). (**A**–**C**) Primary root length (PRL), total lateral root length (TLRL), and *m* values for accession Col-0 with supplemented ABA or phosphorous (P) deficiency. (**D**–**F**) Comparison between PRL, TLRL, and *m* values between wild type (WT, Col-0) and the inorganic phosphate transport mutant *pho1*. Letters indicate Tukey-HSD groups (*p* < 0.05) from a two-way ANOVA with interaction for P and ABA effects. Asterisks indicate significance level obtained from a two-tailed student test: * *p* < 0.05, *** *p* < 0.001.

**Table 1 plants-08-00218-t001:** Two-tailed Pearson correlation coefficients between rhizochron index (*m*) and other conventionally measured traits. TRL = total root length, PRL = primary root length, TLRL = total lateral root length, RD = root density (^LRn^/_PRL_), TLr = ^TLRL^/_TRL_ ratio, PLr = ^TLRL^/_PRL_ ratio, BZ = PR branching zone length, pre- and post-BZ = PR lengths before and after BZ.

Trait	Day 8		Day 10		Day 12		All Stages	
	*p*	*r* ^2^	*p*	*r* ^2^	*p*	*r* ^2^	*p*	*r* ^2^
TRL	0.045	0.015	0.028	0.016	0.006	0.024	0.083	0.003
PRL	0.022	0.020	0.152	0.007	0.002	0.030	0.001	0.012
TLRL	<0.001	0.083	<0.001	0.042	<0.001	0.053	<0.001	0.016
RD	0.250	0.005	0.568	0.001	0.075	0.010	0.763	<0.001
TLr	<0.001	0.149	<0.001	0.078	<0.001	0.088	<0.001	0.056
PLr	<0.001	0.137	<0.001	0.069	<0.001	0.075	<0.001	0.041
BZ	0.056	0.014	0.340	0.003	<0.001	0.047	0.003	0.010
pre-BZ	<0.001	0.050	0.030	0.016	0.005	0.025	<0.001	0.024
post-BZ	0.844	<0.001	0.767	<0.001	0.227	0.005	0.743	<0.001

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
