# Peer review of "A Linear Model to Describe Branching and Allometry in Root Architecture"

_plants, 2019, doi:10.3390/plants8070218_

Round 1
Reviewer 1 Report
This is a review report of the revised manuscript ‘A linear model to describe branching and allometry in root architecture’.
Compared with the previous version, the Authors 1) widened assessment of the m index applicability of the data from earlier study on root systems of mutant plants and plants grown in P and ABA deficiency (I guess that the data concern Arabidopsis plants, although the species is not mentioned), 2) introduced technical corrections like enlarging plots and removing letters from the figures, and 3) indicated possible advantages of applying the m index in allometric study. However, in my opinion it is not all to be done to significantly improve the manuscript.
In their response to my previous comments the Authors declare: our manuscript is a methods paper. If so, it should be clearly stated at the beginning of the paper. Otherwise, the text should be reorganized to a form of a typical research article with separate sections Introduction, Methods, Results and Discussion including a section-appropriate content. From my point of view, it is lack of the mentioned arrangement that disturbs the flow of the story and the clearness of the text because suddenly, out of the blue, you find description of a method in the section Results!
Below, there are two fragments of the Authors’ response to the reviewer comments:
We are not suggesting our model is a profound revolution on RAS phenotyping, but
rather a useful bidimensional parameter that can distinguish between accessions and
environments through the PR and LR elongation allometric relationship, and therefore, m could
correspond to a new root cryptotype: a composite phenotype that is actually subject to natural
selection, but has been overlooked.
We agree with the reviewer that the negative correlation is rather obvious, but we
were also surprised that little attention has being paid to it. This is why we decided to quantify it
and test its suitability to describe root architecture. We now present a new analysis to show its
usefulness on evaluating nutrient deficiency, hormonal effects, and mutant analysis. Our current
work is also testing the m index in other species, but that will be the subject of another
manuscript
First of the fragments indicates possible applications of the m index. The second fragment clearly presents the purpose of the study. Rewritten in the relevant form, the fragments should be incorporated into Discussion and Introduction, respectively.
Introduction still lacks a brief description of the root system development in Arabidopsis. Such a description is needed to enable the reader understanding differences in the root architecture between various accessions.
Abbreviations should be explained where they are first introduced (Abstract line 6; PR, LR).
In Abstract the term ‘concept’ in relation to the root system was replaced with ‘structure’. However, in Introduction ‘concept’ appears again in this context (first line).

Author Response
REVIEWER: This is a review report of the revised manuscript ‘A linear model to describe branching and allometry in root architecture’.
Compared with the previous version, the Authors 1) widened assessment of the m index applicability of the data from earlier study on root systems of mutant plants and plants grown in P and ABA deficiency (I guess that the data concern Arabidopsis plants, although the species is not mentioned), 2) introduced technical corrections like enlarging plots and removing letters from the figures, and 3) indicated possible advantages of applying the mindex in allometric study. However, in my opinion it is not all to be done to significantly improve the manuscript.
RESPONSE: We have added the name of the species A. thalianato the corresponding experiments.
REVIEWER: In their response to my previous comments the Authors declare: our manuscript is a methods paper. If so, it should be clearly stated at the beginning of the paper. Otherwise, the text should be reorganized to a form of a typical research article with separate sections Introduction, Methods, Results and Discussion including a section-appropriate content. From my point of view, it is lack of the mentioned arrangement that disturbs the flow of the story and the clearness of the text because suddenly, out of the blue, you find description of a method in the section Results!
RESPONSE: We now mention in the third sentence of the abstract that our manuscript is a methods paper. Also, the last paragraph of the introduction now mentions that we propose a new method.
As for the arrangement of the manuscript, we have used a similar narrative in a previous paper (Ristova, Rosas, et al, 2013, Plant Physiology; https://doi.org/10.1104/pp.112.210872). Other authors have also adopted a similar writing style in a methods paper (Passot, et al, 2018, Plant Physiology,doi: 10.1104/pp.17.01648). Plantshas already published a manuscript in the Plant Modeling section, which does not have Materials and Methods, but all that section is embedded in the Results. We truly believe that the reviewer intends for us to improve our manuscript, however, we do consider that the manuscript organization works well. This has been the opinion of some of our colleagues who have read the manuscript. We apologize to the reviewer for not further altering the manuscript organization.
REVIEWER: Below, there are two fragments of the Authors’ response to the reviewer comments:
We are not suggesting our model is a profound revolution on RAS phenotyping, but
rather a useful bidimensional parameter that can distinguish between accessions and environments through the PR and LR elongation allometric relationship, and therefore, mcould correspond to a new root cryptotype: a composite phenotype that is actually subject to natural selection, but has been overlooked.
We agree with the reviewer that the negative correlation is rather obvious, but we
were also surprised that little attention has being paid to it. This is why we decided to quantify it and test its suitability to describe root architecture. We now present a new analysis to show its usefulness on evaluating nutrient deficiency, hormonal effects, and mutant analysis. Our current work is also testing the mindex in other species, but that will be the subject of another manuscript
First of the fragments indicates possible applications of the m index. The second fragment clearly presents the purpose of the study. Rewritten in the relevant form, the fragments should be incorporated into Discussion and Introduction, respectively.
RESPONSE: We have included both paragraphs to the suggested sections of the manuscript.
REVIEWER: Introduction still lacks a brief description of the root system development in Arabidopsis. Such a description is needed to enable the reader understanding differences in the root architecture between various accessions.
RESPONSE: We added a paragraph in the introduction where we explain the origins of the root branching pattern.
REVIEWER: Abbreviations should be explained where they are first introduced (Abstract line 6; PR, LR).
RESPONSE: Every abbreviation is spelled out the first time is used in the text.
REVIEWER: In Abstract the term ‘concept’ in relation to the root system was replaced with ‘structure’. However, in Introduction ‘concept’ appears again in this context (first line).
RESPONSE: We apologize for the omission. We have corrected the term.
Reviewer 2 Report
The improved version of the MS is now suitable for the publication.
Author Response
We thank the reviewer for his/her time and inputs to improve the manuscript.